# Constructability Criteria for Farmland Reclamation and Vegetable Cultivation Using Micro-Dam Sediments in Tigray, Ethiopia

**Kazuhisa Koda** [1],*, **Gebreyohannes Girmay** [2] and **Tesfay Berihu** [2]

[1]  Rural Development Division, Japan International Research Center for Agricultural Sciences, Ibaraki 3058686, Japan

[2]  College of Dryland Agriculture and Natural Resources, Mekelle University, Mekelle 231, Ethiopia; gebreyohannes.girmay@gmail.com (G.G.); tesra@yahoo.com (T.B.)

*  Correspondence: kodakazu@affrc.go.jp; Tel.: +81-29-8386676

**Abstract:** The Ethiopian agriculture sector is characterized by rain-fed smallholder systems. The Ethiopian Government has promoted micro-dam construction in micro-watershed in Tigray for the past two decades. The lack of proper conservation measures to control severe soil erosion at the micro-watershed level, however, has often filled downstream micro-dams in with sediments. Sedimentation has affected the irrigation performance of micro-dams due to their bottom pipes becoming clogged with nutrient-rich soils eroded from upstream farmlands. While there is a growing need for adequate resource management to mitigate severe soil erosion at the watershed-level, it is urgent that methods to make use of the sediments deposited in micro-dam reservoirs to facilitate rural agricultural development are discovered. One practical solution is to use sediments to rehabilitate the bare land excavated for micro-dam embankment construction and turn it into reclaimed farmland. The purpose of this paper is to relate the constructability criteria to the farmland reclamation to solve sedimentation problems. This case study reports the yield of vegetable cultivation on farmland reclaimed using sediments from a micro-dam reservoir in Tigray. This case study highlights the practical potential of such a method to contribute to the livelihoods of farmers through the production of vegetable cash crops. The future research needs cost reduction factors on durability, safety or other related aspects to improve our "Constructability Criteria" approach.

**Keywords:** micro-dam; sedimentation; reclaimed farmland; constructability; Ethiopian highlands

## 1. Introduction

Ethiopia is a Sub-Saharan African country that covers an area of 109.7 million km$^2$. Its climate is classified as tropical monsoon due to the low latitude location, and weather varies depending on the topographic elevation. During intense rainfall events, surface runoff flows down slopes and carves deep V-shaped valleys. About 45% of the land in Ethiopia is located at elevations greater than 1500 m above sea level. [1]. The Ethiopian agriculture sector has continuously suffered from poor soil nutrients levels caused by long-lasting cultivation and accompanying soil losses in the form of sheet, rill, and gully erosion [2]. Soil erosion in the Ethiopian highlands is particularly severe due to the sparse vegetation cover. Woldearegay et al. [3] found that the region loses nutrient-rich topsoil at the rate of over 130 tons ha$^{-1}$ year$^{-1}$. Although farmers in the Ethiopian highlands cultivate wheat as a staple food, the wheat growth period and the wet season coincide. Especially in the northern Ethiopian highlands, soil loss can cause sudden great damage, and effective control of this is difficult [4–7]. Sedimentary, igneous, and metamorphic rocks are found in Tigray [8]. The micro-watershed with steep slopes is subject to soil erosion, which is driven by conventional tillage practices.

Ethiopia's main industry is agriculture with cereals (wheat, barley, and indigenous teff, etc.), pulses (bean and chickpea, etc.), and coffee. The agriculture, forestry, and fishing sector accounts for 31% of GDP, the agriculture sector accounts for 65% of employment in 2018, and the agricultural land area accounts for 36% of the total land [9]. Agricultural development is needed in Ethiopia. While there is a great demand for food for the increasing population, Ethiopia's food security is severely undermined by soil nutrient loss. While sediments are deposited in micro-dam reservoirs, watershed lose a considerable quantity of soil nutrients by losing sediments to micro-dams. Sediment-fixed nutrient export due to soil erosion is a severe nutrient loss process that contributes to soil degradation [10]. The sediments in micro-dams with a high nutrient accumulation are left unused, while degraded areas, which used to be farmlands, have been abandoned, despite the increasing demand for higher-yield farmland to produce food for the growing population [10]. The storage volume and lifetime of the micro-dams are reduced because the soil erosion dumps sediments in downstream micro-dams [3,5]. The effects of fast population growth, dry weather, and the small area of arable farmland have extended desertification and land deterioration. These problems need to be mitigated because more than 60% of people in the Ethiopian highlands work in the agriculture sector.

In the Ethiopian highlands, poor water resources significantly affect the yield of agricultural crops. Most farming areas in Ethiopia are rain-fed and are therefore vulnerable to the highly variable rainfall distribution. Growth of the agriculture sector is affected by droughts, which occur once every 2.4 years. The effect has been very severe, especially in northern Ethiopia [11]. To mitigate water shortages in efforts to cultivate more crops for the increasing population, the Ethiopian government has, for more than two decades, constructed water-harvesting facilities, including micro-dams [5,8,10]. The construction of micro-dams was planned to achieve various economic, hydrologic, and ecological goals, including increased food production, easy access to available drinking water for people as well as livestock, a rise in the groundwater level, and the emergence of new springs [5]. A micro-dam is a small dam or reservoir to store water for domestic, livestock, and irrigation purposes. It has been estimated that 50% of the micro-dams in northern Ethiopia have seen their life expectancy decline from 26 to 13 years due to sedimentation [10–12]. Sedimentation has resulted in lower levels of soil nutrients, such as organic carbon (OC), total nitrogen (N), available phosphorus (P), and exchangeable cations, being found in the soils of the watershed than in micro-dam sediments [10].

The Ethiopian government, in collaboration with international organizations, embarked on a large-scale soil and water conservation strategy through the construction of micro-dams in Tigray; indeed, more than 50 micro-dams were built in the region from 1996 to 2001 [5,13]. Due to a lack of good planning, however, including the selection of appropriate micro-dam sites and technologies, these micro-dams suffered from serious sedimentation and water leakage, resulting in the failure of expected functions [1,12,14,15]. Based on a study conducted in Tigray on micro-dam sedimentation in relation to soil erosion in the watershed, Tamene et al. [5] reported that most of the micro-dams constructed to harvest rainwater lost 50% of their storage capacity less than five years after becoming operational. Haregeweyn et al. [12] showed that 50% of the 13 studied micro-dams had lost half of their life expectancy, while only three micro-dams were estimated to operate for their total expected lifespan [16]. Furthermore, based on an analysis of 92 micro-dams, Berhane et al. [8] found that 61% had sedimentation/siltation problems, 53% suffered from leakage, 22% experienced insufficient inflow, 25% had structural damage, and 21% had spillway erosion problems. Rapid sedimentation is mainly caused by poor planning of the micro-dams [12].

The purpose of this study was to demonstrate the utility of constructability criteria for reclaimed farmland to mitigate the sediment accumulation in micro-dams. This study showed the vegetable cultivation on reclaimed farmland using sediments from the micro-dam reservoir. It was found that micro-dam sediments consisted of fine clay and some were suitable for farmland reclamation on bare land [10,16]. However, the erosion process and the sedimentation usage method in the micro-dam has not been well studied, so onion was cultivated on reclaimed farmland by using the sediments and its yield was compared against the Ethiopian average.

## 2. Materials and Methods

### 2.1. Study Site

The Tigray region is situated in the northern Ethiopian highlands. The climate is characterized as tropical alpine and semi-arid. The topography of the region mainly consists of highland plateaus up to 3900 m a.s.l. (above sea level), which are divided by gorges [12]. In terms of geohydrology, the region is dominated by different types of rocks and soils [3]. The rough topography with rocky geohydrology is very sensitive to erosion, making effective utilization and management measures important. The highland climate has sustained a high population density with a long cultivation history, which is estimated to date back to 3000 BCE [12]. The long-term use of farmlands for crop cultivation, combined with the steep topography, water erosion, and insufficient vegetation cover caused by cutting almost all of the residues of wheat, teff, and barley, etc., has caused serious land degradation. Consequently, the Tigray region is considered to be one of the most degraded (and still degrading) regions in Ethiopia [3,5]. The land degradation, coupled with abnormal rainfall distribution, has caused recurring drought and famine, which was historically demonstrated during 1888–1892, 1973–1974, and 1984–1985 [13].

This research was conducted from 2017 to 2020 in the Adizaboy micro-watershed, which covers an area of about 8.5 km$^2$ and is located between latitudes of 13.64° N and 13.68° N, and between longitudes of 39.56° E and 39.6° E (Figure 1). Adizaboy micro-watershed ranges in altitude from 2050 to 2275 m a.s.l. The annual rainfall in Wukuro city near this area varies from about 300 to 1000 mm. Slope survey results along the representative survey line (Figure 1c) in the Adizaboy micro-watershed show that the average slope of the survey line was 8.8%, which is classified as a steep slope.

The Adizaboy micro-dam is situated at the exit of the Adizaboy micro-watershed (Figure 2). The Adizaboy micro-dam was built in 2009 with local soils and rocks taken from an upland field near the dam site. It is labeled as a zoned rockfill type, which has a crest height of 9.06 m from the bottom pipe and is somewhat smaller than the average of other micro-dams in Tigray [8]. The water surface of the Adizaboy micro-dam seasonally changes. It sometimes becomes dry in March–April. The upstream slope of the Adizaboy micro-dam embankment (H3: V1) is gentler than the downstream slope (H2.55: V1). The trail for farmers in the micro-watershed traverses in a north–south direction over the embankment. The reclaimed farmland that used the micro-dam sediments is situated near the Adizaboy micro-dam and is above the micro-dam crest.

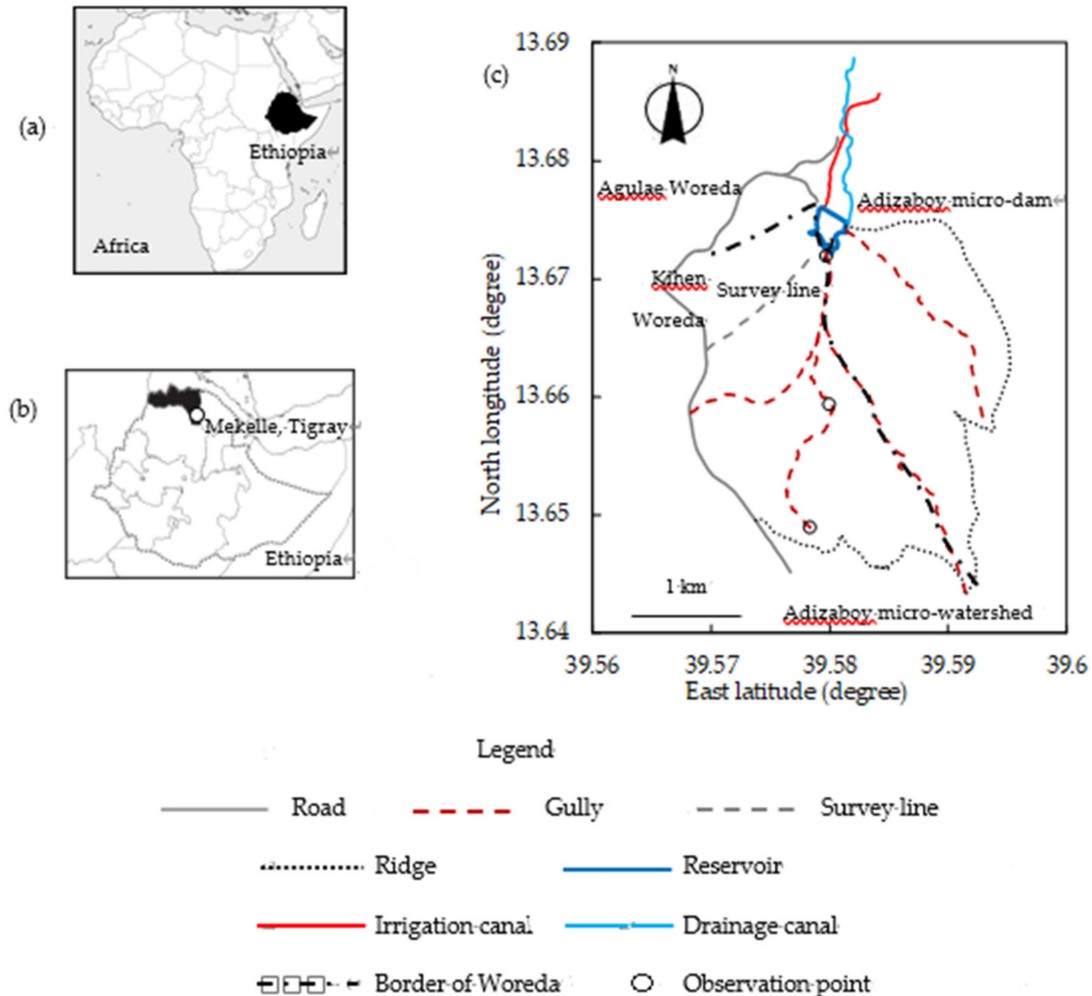

**Figure 1.** Location map of the Adizaboy micro-watershed in Tigray: (**a**) Location map of Ethiopia in Africa (black area displays Ethiopia); (**b**) location map of Mekelle, Tigray in Ethiopia (bold dotted line shows the national boundary of Ethiopia, and the black area is Tigray); and (**c**) overview of Adizaboy micro-watershed.

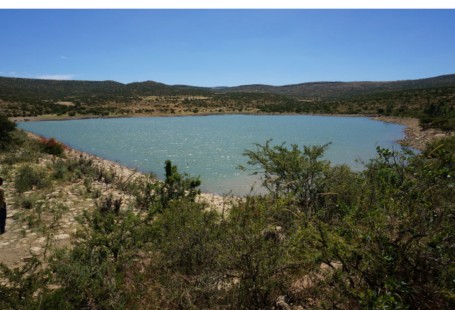

**Figure 2.** Adizaboy micro-dam (photo aspect, NW–SE).

### 2.2. Gullies in the Micro-Watershed

Some gullies developed in the upper reaches of the micro-watershed, which caused sediment accumulation in the Adizaboy micro-dam. The uncovered soil mainly included Cambisols, which is suitable for agriculture [17]. In the upstream and middle reaches of upland areas of the Adizaboy micro-watershed wheat is mainly cultivated by rainwater. There is about 1.82 km$^2$ of cropland in the

area of the Adizaboy micro-watershed, accounting for 21.4% of the total; 87.1% of the cropland is situated on comparatively flat land with a slope of less than 10° [18].

The geological layers in the Adizaboy micro-watershed consist of weathered marl, shale, and limestone [15]. The rainfall and water flow in the micro-watershed caused fast outflow of fine particles and slow outflow of coarse particles. The location of the three observation points is shown in Figure 1c.

1). Upstream

Four major gullies formed in the upper reaches of the Adizaboy micro-watershed as a result of surface outflow and underground infiltration caused by rainfall. At the starting point of the gully, the depth of the surface collapses and a valley-shaped topography formed of about 2 to 2.5 m height (Figure 3). At this point, water entered through rill erosion due to intense rainfall. Even after rainfall stopped, permeated water continued to enter the gully erosion channel. The starting point of gully erosion then gradually extended upwards.

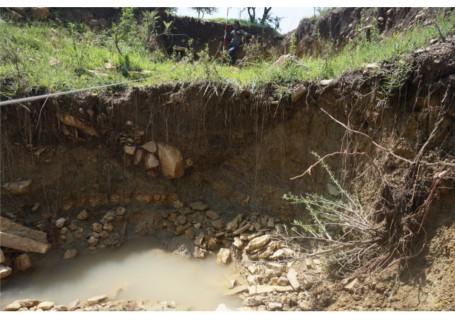

**Figure 3.** The starting point of gully erosion in the upper reaches of the Adizaboy micro-watershed (photo aspect, NE–SW).

Figure 4 shows gully erosion in the upper reaches at the observation point. The shape of the cross-sectional views of the gully was deeper around the outside part because the water velocity was relatively faster there. Sand, stone, and gravel collapsed at the slopes of the gullies and this material was transported downstream. Inflows of sand and stone were greater than the outflows in the gully from the starting point to 60 m, although outflows were larger than inflows in other parts. The soil erosion volume depended on the distance from the starting point of the gully because of the slope collapse and water velocity effect.

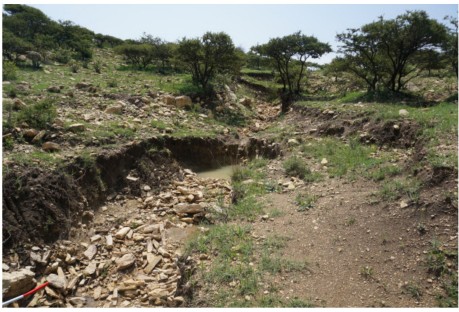

**Figure 4.** Gully erosion upstream of the Adizaboy micro-watershed (photo aspect, NE–SW).

2). Middle reaches (Figure 5)

The water velocity was higher in the middle reaches of the Adizaboy micro-watershed and the depth of gully erosion was about 1.0–1.5 m. Gullies in the middle reaches were slightly wider than in the upper reaches. The slope in the middle reaches was often steeper than the one in the upstream, and many boulders existed as well. The widths were sometimes very narrow and, at depths of more than 2 m, the soft geological layers were eroded.

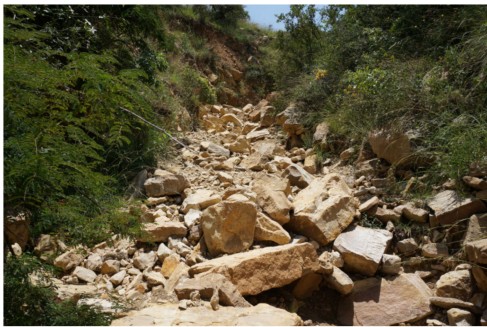

**Figure 5.** Gully erosion in the middle reaches of the Adizaboy micro-watershed (photo aspect, S–N).

3). Near micro-dam

The gully near the micro-dam is gently sloped (Figure 6) but wider, with a width from 8 to 10 m. The depth is under 0.5 m. The shape of gravel and stone transported is rounder and smaller but there are still some big pieces of stone with a diameter over 0.5 m. As sand, silt and clay are smaller and lighter, they are transported to the micro-dam. Some of them go down-stream in the irrigation canal through the bottom pipe, and some of them stay in the micro-dam and accumulate at the bottom of the micro-dam.

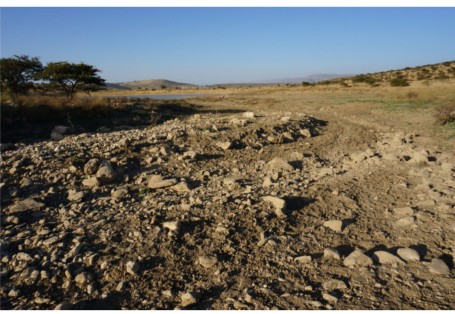

**Figure 6.** Gully erosion near Adizaboy micro-dam (photo aspect, S–N).

*2.3. Sedimentation in Micro-Dam and Reclaimed Farmland*

The new bathymetric survey method (topographic survey) was applied to estimate the sediment volume for a short period [19]. An echo-sounder was used to measure the sediment surface depth from a boat. The coordinates around the perimeter of the micro-dam and the water depth were recorded (Figure 7). The sediment volume between the measured sediment surface and the estimated bottom was calculated to be 6400 m$^3$. Table 1 shows the sediment characteristics versus the bare land.

Micro-dam sediments were transported to the reclaimed farmland in March and April 2017 when the surface water of the Adizaboy micro-dam disappeared in the dry season. The construction period was about 2 weeks. The micro-dam sediment was excavated manually and transported by donkey to the farmland. The reclaimed farmland was constructed in the following sequence: 1) stone bunds were installed around the perimeter of the reclaimed farmland, with a height of 0.5 m and width of 0.6 m; 2) sediments were transported and leveled in the farmland; and 3) a diversion canal was constructed on the ground around the reclaimed farmland. The farmland reclamation was conducted on bare land about 100 m from the Adizaboy micro-dam. The dimensions of the farmland were 23 × 14 m (322 m$^2$).

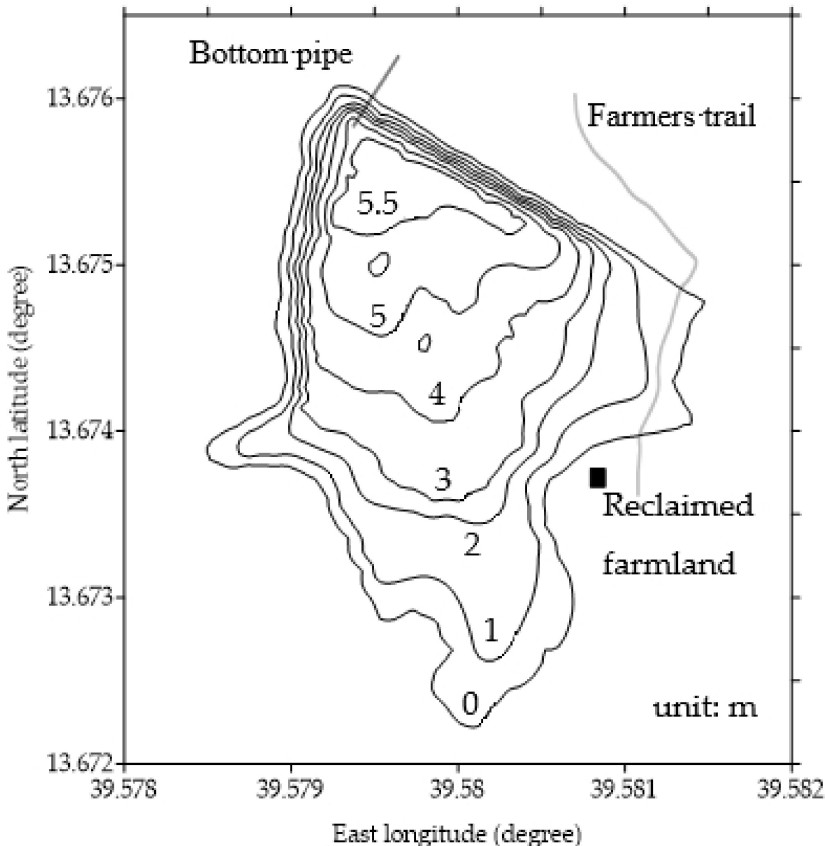

**Figure 7.** Sediment survey results for Adizaboy micro-dam. The micro-dam depth contour map was measured with an echo-sounder; numerical characters show the water depth (m) compared to the full water level.

**Table 1.** Micro-dam sediment and bare land soil sample characteristics [19].

| Items | Sediments | Bare Land |
|---|---|---|
| Sand (g/kg) | 70 | 102 |
| Silt (g/kg) | 536 | 501 |
| Clay (g/kg) | 394 | 397 |
| Bulk density (g/cm$^3$) | 1.1 | 1.2 |
| Field capacity (v/v%) | 28.8 | 15 |
| Permanent wilting point (v/v%) | 13.3 | 9.2 |
| Available water capacity (v/v%) | 15.6 | 5.9 |
| Available water capacity (mm/15cm depth) | 23.3 | 8.8 |
| pH (H$_2$O) | 7.3 | 8.1 |
| Organic carbon (g/kg) | 24.7 | 17.6 |
| Total nitrogen (g/kg) | 3.3 | 3.4 |
| Available phosphorus (mg/kg) | 9.2 | 8.8 |
| Exchangeable potassium (cmol(+)/kg) | 25.0 | 14.1 |

## 2.4. Constructability Concepts

The Construction Industry Institute (CII) Constructability Task Force defines constructability as the "optimum use of construction knowledge and experience in planning, design, procurement, and field operations to achieve overall project objectives" [20].

A general framework to be covered with constructability research was worked out by Vanegas [21]. Research from the CII established 17 constructability concepts, and the CII built up a Constructability Concepts File [22] that describes useful examples regarding the application of each concept. These

constructability concepts were classified into three major project delivery phases: 1) conceptual planning, 2) design and procurement, and 3) field operations. The CII also published a Constructability Implementation Guide [23], which outlines a system of methods through which to apply constructability. Table 2 provides the constructability concepts.

**Table 2.** Constructability concepts.

| **Conceptual Planning Phase** |
| --- |
| Concept 1-A: The constructability program should be made an integral part of the project execution plan. |
| Concept 1-B: Special emphasis should be placed on maintaining an effective project team. |
| Concept 1-C: Early project planning should actively involve individuals with current construction knowledge and experience. |
| Concept 1-D: This early construction involvement should be a consideration in developing the contracting strategy. |
| Concept 1-E: The master project schedule should be start-up and construction-sensitive. |
| Concept 1-F: Major construction methods should be analyzed in-depth early on and should be facilitated through proper facility design. |
| Concept 1-G: Site layouts should promote efficient construction, operation, and maintenance. |
| **Design and Procurement Phase** |
| Concept 2-A: Design and procurement schedules should be construction-driven. |
| Concept 2-B: The capabilities and benefits of advanced information technology should be exploited. |
| Concept 2-C: Designs should be configured to enable efficient construction. |
| Concept 2-D: Design elements should be standardized. |
| Concept 2-E: Technical specifications should promote construction efficiency. |
| Concept 2-F: Detailed designs of modules and preassemblies should be prepared to facilitate efficient fabrication, transport, and installation. |
| Concept 2-G: Project designs should promote accessibility to materials and equipment by construction personnel. |
| Concept 2-H: Designs should allow for and enable construction under adverse weather conditions. |
| **Field Operation Phase** |
| Concept 3-A: Special effort should be applied toward developing innovative construction methods. |

## 3. Results

### 3.1. Vegetable Cultivation on Reclaimed Farmland

Onion (*Allium cepa*) cultivation began in January 2019 and harvest took place in June 2019. We believed onions should grow with a sediment depth of about 20 cm, which included soil loss of 5 cm, in terms of cost/benefit ratio [10]. The shallow-rooted onion was selected as it does not require a deep cover of soil. Furrow and drip irrigation methods were adopted. Irrigation water was taken from the Adizaboy micro-dam and was stored in a concrete farm pond located at the highest point in the reclaimed farmland site (capacity of 85 m$^3$). The two irrigation methods for onion planting had 12 replications each and were grown at 25 cm intervals in rows, with 10 cm between plants. The irrigation requirement was 17.5 mm/day.

Figure 8 shows the onion yield on reclaimed farmland. Micro-dam sediments are classified into cohesive soil, which holds nutrients, fertilizer, and water, but does not have good air permeability. Abundant irrigation water naturally moved out through the stone bund, which surrounded the perimeter of the reclaimed farmland. The drainage problem did not occur on reclamation farmland. According to the Tukey–Kramer method (multiple comparison procedure for statistical analysis),

the onion yield on reclaimed farmland with drip irrigation was significantly higher than that on reclaimed farmland with furrow irrigation and that of the national average (2006 to 2017).

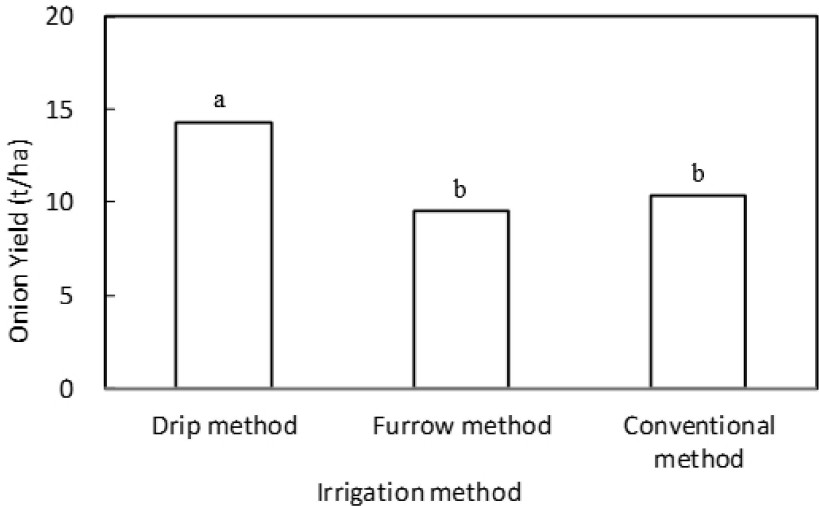

**Figure 8.** Onion yield on reclaimed farmland for different methods. Letters above onion yields show significant differences ($p < 0.01$) among treatments.

### 3.2. Constructability Criteria

Farmland reclamation processes included the construction of 1) a stone bund; 2) sediment layers; 3) a drainage canal; 4) a weather observation device; 5) irrigation facilities such as a pump, a hose, a farm pond, and a water tank; 6) a warehouse to store equipment, as well as to accommodate a guard; and 7) a barbed wire fence with prickly timber attached to metal columns to protect agricultural products against attacks by wild animals when deemed necessary, depending on local conditions.

After the farmland site was selected, weeding, shrub-clearing, and the removal of large stones began. Stones were utilized to construct the bunds along the boundaries of the farmland. Micro-dam sediments were collected by shovels and transported by donkeys to the farmland. Transferred sediments were layered and leveled on the reclaimed farmland, from which small stones and weeds were removed, so that the farmland surface was made flat and conducive to farming. Before seeding on the farmland, fences with barbed wire were established to keep away domestic animals such as goats, sheep, and cattle, which might have grazed the farmland, destroying planted crops. A warehouse was built to accommodate a guard to watch the crops and prevent theft or damage by wild animals, as well as to store equipment such as water pumps and drip irrigation. After water flows from upstream catchments to the reclaimed farmland were observed during rains, a drainage system was established so that the water eventually flowed into the micro-dam without causing erosion problems for the reclaimed farmland.

The constructability criteria and their attributes based on the authors' experience were used to guide the implementation of the farmland reclamation practice for each phase. Each criterion is examined here and the key attributes are summarized. We participated in the stakeholder meeting held at the public hall in Agulae Woreda on June 1, 2019. The meeting was attended by the Ministry of Agriculture and Rural Development in Tigray, Mekelle University, and about 50 farmers. After the meeting was over, all participants moved to the reclaimed farmland site and we explained our activities to them. Our guidance for the technical feasibility and acceptance by the community and other stakeholders was validated. These factors had a large effect on the up-scalability of the approach.

### 3.2.1. Conceptual Planning Phase

(1) Building an effective project team

The success of a farmland reclamation project depends on the capacity of project team members in designing, procurement, and field operations through relevant training, incentives, and communication. Labor productivity could be improved by concrete mixing training. This criterion is related to the constructability concepts 1-B, 1-C, and 1-D. The attributes required are described as follows: 1) a training program for specific crafts, 2) daily allowances for on-site jobs, 3) on-site communication with persons who have construction expertise, 4) on-site teamwork under the leadership of those with construction expertise, 5) availability of delivery systems such as a horse cart and a car, and 6) availability of special craftsmen and equipment for metal welding.

(2) Facilitating proper designs and layouts

Decisions on appropriate construction methods, facility designs, and site-layouts must be based on in-depth analyses to promote efficient construction, operation, and maintenance, utilizing information and survey data. This criterion is related to constructability concepts 1-E, 1-F, and 1-G. The attributes required are as follows: 1) the amount of storage water in the micro-dam required for irrigation, and 2) the availability of standard designs of farm pond, tank, and drip irrigation.

(3) Choosing suitable sites for reclaimed farmland considering the whole implementation planning

The selection of suitable sites for reclaimed farmland comes at the beginning of the implementation planning process (Figure 9). It is especially important for the sustainability of farmland reclamation works to consider engineering factors from a farmer's perspective. For example, the dimensions of reclaimed farmland should be consistent with the farmer's capacity and demand. Distance, as well as height difference, between micro-dams and reclaimed farmland, should be kept to a minimum because it is difficult to transport sediment and to pump irrigation water from micro-dams to reclaimed farmlands if the distance becomes too great. Farmland should not have steep slopes, so as to prevent soil loss and retain the sediment thickness.

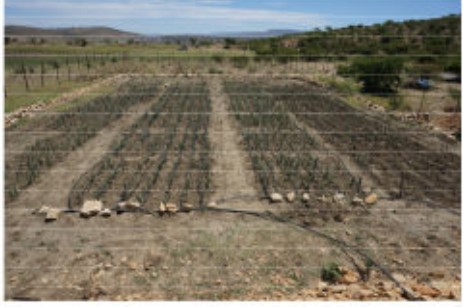

**Figure 9.** Drip and furrow irrigation were conducted on the reclaimed farmland (photo aspect, S–N).

It is also important during site selection to consider the logistics, as farmland reclamation works are often implemented in rural remote areas. The site conditions must be exploited to maximize efficiency. For example, the availability of construction materials (e.g., stone, gravel, and sand) and sediment to be reclaimed in surrounding areas facilitates mobilizing locally available resources for operation. In turn, procuring external construction materials and tools (cement for foundation work, nails, iron bars, galvanized sheets, barbed wire, and wood poles) as well as inputs, such as extra chemical fertilizer to complement the fertility of sediments, requires good accessibility to reclaimed farmlands for easy delivery of supplies. The work area should be large enough to allow on-site activities such as assembly and concrete mixing for the construction of fences, along with space for the construction of a warehouse to store equipment, a guardhouse for protection against wild animals, and a farm pond with tanks to store irrigation water. After the farmland has been reclaimed, accessibility is still critical for the farmers who will be commuting to the farmland to grow crops and deliver crops to market. There should also be sufficient space to expand reclaimed farmland in the future. This criterion is related to

constructability concepts 1-A and 1-G. The following attributes should be taken into consideration: 1) compatibility of reclaimed farmland dimensions with farmer's operability and maintainability; 2) the distance and the height difference between micro-dam and reclaimed farmland; 3) slope of farmland and thickness of sediments; 4) clearing and leveling of reclaimed farmland; 5) fencing the reclaimed farmland; 6) site accessibility for material delivery, as well as for farmers; 7) adequate laydown area availability for requisite working space, as well as a warehouse in which to keep necessary heavy equipment and tools; and 8) extra space for future site expansion.

### 3.2.2. Designing and Procurement Phase

(1) Designing efficient construction elements

When designing farmland, costs should be minimized, including procurement costs and costs for labor, materials, equipment, and guards. This criterion is related to constructability concepts 2-B, 2-C, 2-D, and 2-E. In doing so, the following points must be considered for designing and procuring schedules: 1) to minimize the complexity of design details and reduce the need for overly detailed specifications, 2) to make use of past survey results and water balance analysis results, and 3) to use standard dimensions and sizes for the reclaimed farmland system.

(2) Preparing for preassemblies and logistics

In order to facilitate efficient field operations, detailed designs of modules, including fabrication, transport, and installation of materials and equipment, should be prepared by construction personnel in advance. Procurement schedules must be planned and designed to minimize potential factors that could delay field operations, such as delays in equipment delivery, customs clearance, and permission processes. An inventory of construction and delivery components can help minimize the costs and time involved in on-site and off-site field operations, thus maximizing efficiency. This criterion is related to constructability concepts 2-F and 2-G. The key attributes are described as follows: 1) construction processes involving maximum use of on-site equipment and minimum labor, 2) off-site preassembly of some materials requiring prefabrication (i.e., weather observation devices) and cutting/welding (e.g., construction components, such as L-type metal columns) by skilled labor, 3) plan to maximize the use of the same transportation system for material and equipment delivery, and 4) facilitation of customs inspection for equipment made abroad.

(3) Preparing for adverse weather conditions

Negative effects due to bad weather must be minimized. This criterion is related to constructability concept 2-H. The key attributes are described as follows: 1) reclamation work, such as sediment transportation in micro-dams and concrete work for a farm pond and foundation work to fix poles, should be limited under rainy conditions; 2) site access through submerged farming roads under rainy conditions should be restricted; and 3) temporary storage for weather-sensitive equipment and materials should be provided.

(4) Planning, design, and procurement schedules and flexibility

The interaction and interface of activities must be well managed in the planning, design, and procurement schedules. Applications for farmland reclamation permits should be made at the earliest available opportunity. Schedules should include flexibility to deal with potential factors that could delay field operations and procurement processes. This criterion is related to constructability concept 2-A. The key elements related to this concept are as follows: 1) land permit processes to obtain the reclaimed land, 2) adaptability to withstand unexpected field conditions, such as extremely high run-off volume, dropping or rising groundwater level, and water consumption by people and livestock in the vicinity, and 3) potential delays due to the unavailability of specialized equipment, material, and labor.

### 3.2.3. Field Operation Phase

It is necessary to maximize the use of advanced and innovative technology and construction techniques. This criterion is related to constructability concepts 2-B and 3-A. The following points must

be considered in the operation works: 1) to maximize the use of advanced materials (solar light), and 2) to maximize innovative survey equipment (GPS, note PCs, cameras, weather observation devices, and echo-sounders).

## 4. Discussion and Conclusions

Micro-dams have been constructed to deal with water shortage problems related to producing more crops in Tigray, Ethiopia. However, problems associated with sedimentation have occurred in many micro-dams. This issue precludes their ability to meet their intended performance levels. This paper showed one possible solution to solve the micro-dam sedimentation problem in the Adizaboy micro-dam, by constructing reclaimed farmland using the micro-dam sediments. The constructability concepts could facilitate farmland reclamation.

Constructability criteria, which generally facilitate the quality control, safety management, and schedule management of a project, were evaluated by the cost reduction of the farmland reclamation. The cost of labor (including guardianship), material, equipment, and groundwork was included in the project. The land use permit issued from Agulae Woreda administrative office could justify and formalize the financing of meaningful farmland reclamation for rural development (relating to Section 3.2.1. (1)). If there was no water in the micro-dam, the water stored in the farm pond was used. If there was no water in the farm pond, then the expensive option of arranging for a water tank truck to transport groundwater in Wukro city to the farm pond was required in our research. Drip irrigation was conducted to save water and avoid the expensive option (relating to Section 3.2.1. (2)). It is important that the effect of sedimentation is mitigated by conducting manual excavation and watershed management. Sediment excavation and transportation to the reclaimed farmland was carried out by preparing and organizing donkeys and local labor (relating to Section 3.2.1. (3)). Watershed management was carried out using conservation agriculture on wheat farmland by wheat residue to protect the farmland from water erosion. Machine excavation and the lost storage replacement were not conducted because they were prohibitively expensive for farmers, although the farming road could have been used for machine transportation.

Table 3 shows the cost reductions made by design change and construction material recycling. Land rent fees for the reclaimed farmland and the use of water and sediments in the micro-dam were waived after we obtained the land-use permit (relating to Section 3.2.2. (4)). Once the reclaimed farmland was handed over to the farmers from the project, they did not have to pay for the guardianship costs as long as they stayed near the reclaimed farmland (relating to Section 3.2.3.). The future research needs cost reduction factors on durability, safety or other related aspects to improve our "Constructability Criteria" approach.

**Table 3.** Cost reduction by design change and construction material recycling.

| Item | Original Design | Modified Design | Rough Cost Estimation | Relation |
|---|---|---|---|---|
| Farm pond | Concrete stairs | Wooden ladder | 12,000 JPY | 3.2.2 (1) |
| Warehouse | New corrugated metal plate | Used corrugated metal plate | 8000 JPY | 3.2.2 (2) |
| Farm pond cover | Use of eucalyptus wood | Reduced use of eucalyptus wood | 1000 JPY | 3.2.2 (3) |

We aimed to mitigate soil erosion problems in a micro-watershed. JIRCAS supported the research activities alongside Mekelle University in terms of the international joint research project. There would have been further room for cost-savings associated with farmland reclamation if the work was not restricted for a short period during the dry season. There are many young landless farmers as well as some conflict victims in the project site. It will be necessary to construct more reclaimed farmland to

meet demand in the future. Farmers will conduct rotational vegetable cultivation of crops such as garlic, onion, potato, and carrot.

The authors have mapped the constructability criteria to solve the micro-dam sedimentation problem through the farmland reclamation by making use of micro-dam sediments, which administrative officers, farmers, and researchers have faced in Tigray. The good or bad performance of planning, design, and construction of the reclaimed farmland is decided by participants' experience, knowledge, teamwork, communication, and leadership. The constructability criteria will produce an optimum reclaimed farmland model to make the most of benefits and reduce costs to the minimum, to build the sustainable food production practices, and to increase the agricultural productivity and incomes of small-scale farmers.

**Author Contributions:** Conceptualization, K.K. and G.G.; methodology, K.K. and G.G.; software, K.K.; validation, K.K. and G.G.; formal analysis, K.K.; investigation, K.K., G.G., and T.B.; resources, K.K., G.G., and T.B.; data curation, K.K. and G.G.; writing—original draft preparation, K.K.; writing—review and editing, K.K.; visualization, K.K.; supervision, G.G. and T.B.; project administration, K.K. and G.G. All authors have read and agreed to the published version of the manuscript.

**Funding:** This research received no external funding.

**Acknowledgments:** Japan International Research Center for Agricultural Sciences and Mekelle University implemented the work, which was conducted as part of the African Watershed Management Project in Ethiopia. The authors thank JIRCAS research coordinator Miyuki Iiyama, and JIRCAS project members Keiichi Hayashi and Tomohiro Nishigaki, for commenting on the manuscript. Thanks also to Mekelle University, Ethiopia for providing a base for this joint project.

**Conflicts of Interest:** The authors declare no conflict of interest.

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
