# Peer review of "Constructability Criteria for Farmland Reclamation and Vegetable Cultivation Using Micro-Dam Sediments in Tigray, Ethiopia"

_sustainability, doi:10.3390/su12166388_

Round 1

Reviewer 1 Report

The manuscript “Constructability Criteria for Farmland Reclamation and Vegetable Cultivation by Using Micro-dam Sediments in Tigray, Ethiopia” reports on a project to use sediments accumulating in a dam in Ethiopia to enrich eroded farmland upstream from the dam. More than anything, this manuscript reads as a project report, summarizing the initial problem, what was done, and what the outcome was. To me, the most interesting and useful part of the paper is Section 3.2, which applies the general criteria for a successful construction project, to the specific project being discussed.

As written, the manuscript feels somewhat isolated. In essence, it is a case study, but, beyond pointing out that there are many micro-dams in Ethiopia like the Tigray dam, they do not try to extend the findings from this project to other cases. The manuscript presents a helpful account of what happened in one place, but without a lot of explicit focus on broader extensions.

The style of the writing is somewhat idiosyncratic, especially in terms of diction, preposition use, etc. To me, this is generally not a barrier to understanding. I commented on a few lines or phrases that I had trouble understanding. The decision of whether or not to require more editing to conform to standard English-language usage for science writing is up to the editors.

p. 2

I do not understand what these figures mean: “The ratio of agricultural land area to total land area accounts for 36%, arable land area 15%, permanent cropland area 1%, and forest area 13% [9].”

This sentence marks the first time micro-dams are mentioned in the main text: “It has been estimated that 50% of the micro-dams in northern Ethiopia have had their life expectancy declined from 26 to 13 years due to sedimentation [10-11].” The context for understanding what they are isn’t provided until the next paragraph. I would move this sentence and the next one to be with the introduction to micro-dams. It also might be helpful to add a sentence or two to that paragraph explaining exactly what a micro-dam is. (As someone who doesn’t know, I just assumed that they are small dams, but this doesn’t answer other questions, such as whether they produce electricity, whether they are primarily a means of creating a reservoir, etc.)

This sentence is unclear to me: “Thus, the goals are more often short-term than expected due to serious soil erosion, which provides sediments to downstream micro-dams, leading to reductions in the storage volume and lifetime of the micro-dams [3,5].”

p. 3

What does this mean: “changeful hydraulic properties”?

What does “imprudent vegetation cover” mean?

p. 4

The photos in this paper are very helpful. Could the authors also include a photo of the Adizaboy dam?

I do not know what this means: “The farmers trail in the micro-watershed traverses in a north-south direction over the embankment.”

p. 6-7

The Concepts from the Constructability Implementation Guide should be reformatted as a table. As text, they take up too much room and disrupt the flow of the narrative.

p. 7

What does this mean: “Constructability knowledge, such as sediment depth on reclamation farmland of about 20 cm, was obtained from the literature review [10] in terms of cost/benefit ratio”? That onions should grow in a medium with a depth of 20 cm?

It is unclear what this means: “Stone bud surrounding reclamation farmland played a role in natural drainage.”

What is the Tukey Cramer method?

p. 11

What does this mean: “The design change of the farm pond was conducted to remove the concrete up-downstairs in it and to reduce the use of the framework, sand, and cement,”?

Author Response

July 17, 2020

To: The Reviewer 1, Sustainability

Dear Sir/Madam;

We would like to submit our revised manuscript entitled “Constructability Criteria for Farmland Reclamation and Vegetable Cultivation by Using Micro-dam Sediments in Tigray, Ethiopia,” which was originally submitted on June 29, 2020, and which has the following manuscript ID “sustainability-865755.”

We thank the Referee 1 for making time available to this paper. Please find the following point-to-point responses to the reviewers’ comments, and attached revised manuscript with changes shown. We would like to thank the reviewer 1 for their valuable comments, which we think have resulted in an improved manuscript. We trust that the revised manuscript will meet the approval of the editor and reviewers, and that it would be suitable for publication in Sustainability.

Sincerely yours,

Kazuhisa KODA

Senior Researcher

Rural Development Division

Japan International Research Center for Agricultural Sciences

Ohwashi, Tsukuba, Ibaraki 305-8686, Japan

Phone : +81-29-838-6676 Fax : +81-29-838-6682

E-mail : <kodakazu@affrc.go.jp>

Response to Reviewer 1 Comments

(Point) The manuscript “Constructability Criteria for Farmland Reclamation and Vegetable Cultivation by Using Micro-dam Sediments in Tigray, Ethiopia” reports on a project to use sediments accumulating in a dam in Ethiopia to enrich eroded farmland upstream from the dam. More than anything, this manuscript reads as a project report, summarizing the initial problem, what was done, and what the outcome was. To me, the most interesting and useful part of the paper is Section 3.2, which applies the general criteria for a successful construction project, to the specific project being discussed.

As written, the manuscript feels somewhat isolated. In essence, it is a case study, but, beyond pointing out that there are many micro-dams in Ethiopia like the Tigray dam, they do not try to extend the findings from this project to other cases. The manuscript presents a helpful account of what happened in one place, but without a lot of explicit focus on broader extensions.

The style of the writing is somewhat idiosyncratic, especially in terms of diction, preposition use, etc. To me, this is generally not a barrier to understanding. I commented on a few lines or phrases that I had trouble understanding. The decision of whether or not to require more editing to conform to standard English-language usage for science writing is up to the editors.

(Response) We would like to express our appreciation to the Reviewer 1 for his or her insightful comments, which have helped us significantly improve the paper “Constructability Criteria for Farmland Reclamation and Vegetable Cultivation by Using Micro-dam Sediments in Tigray, Ethiopia.”

The problem we have challenged in the present paper will be a major issue in the field. We appreciate the Reviewer 1’s comment. However, we think that we have chosen one of the representative micro-dams. In general, the sedimentation problem in Adizaboy seems to be very similar to that in other micro-dams based on the literature review. Unfortunately, we have not been able to go out of Tigray region due to the instability of the political situation in Ethiopia. Our field activities, which have recently been conducted, included other micro-dams such as Maileba, Gum 30, Shilanat, and Ground Ho in Tigray, etc. We had a meeting with a JICA expert in Ethiopia several times in ADD as well. We were about to apply for extending our training activities in cooperation with JICA and Embassy of Japan in Ethiopia as well as to prepare for the electrical prospecting to specify the water leakage location of Adizaboy micro-dam for the further study just before the spread of COVID-19.

Point 1: p. 2 I do not understand what these figures mean: “The ratio of agricultural land area to total land area accounts for 36%, arable land area 15%, permanent cropland area 1%, and forest area 13% [9].”

Response 1: P.2 L.1

We appreciate the Reviewer’s useful comments on the statistical data. I think that these figures mean that “It will be necessary to conduct agricultural development in Ethiopia,” which has been added after “The ratio of agricultural land area to total land area accounts for 36%, arable land area 15%, permanent cropland area 1%, and forest area 13% [9].”

Point 2: This sentence marks the first time micro-dams are mentioned in the main text: “It has been estimated that 50% of the micro-dams in northern Ethiopia have had their life expectancy declined from 26 to 13 years due to sedimentation [10-11].” The context for understanding what they are isn’t provided until the next paragraph. I would move this sentence and the next one to be with the introduction to micro-dams. It also might be helpful to add a sentence or two to that paragraph explaining exactly what a micro-dam is. (As someone who doesn’t know, I just assumed that they are small dams, but this doesn’t answer other questions, such as whether they produce electricity, whether they are primarily a means of creating a reservoir, etc.)

Response 2: P.2 L.7

Thank you for pointing out this error. In view of the Reviewer’s comment, we have added the following sentence.

“A micro-dam is a small dam or reservoir to store water for domestic, livestock, and irrigation uses.”

The following two sentences have also moved to the next paragraph.

“It has been estimated that 50% of the micro-dams in northern Ethiopia have had their life expectancy declined from 26 to 13 years due to sedimentation [10-11]. Such a process resulted in lower soil nutrients, such as organic carbon (OC), total nitrogen (N), available phosphorus (P), and exchangeable cations in the soils of the watershed than those in micro-dam sediments [10].”

Reference numbers have been changed from 11 to 12 and from 12 to 11.

Point 3: This sentence is unclear to me: “Thus, the goals are more often short-term than expected due to serious soil erosion, which provides sediments to downstream micro-dams, leading to reductions in the storage volume and lifetime of the micro-dams [3,5].”

Response 3: P.2 L.14

In view of the Reviewer’s comment, we have revised the following sentence.

“Thus, the goals are more often short-term than expected due to serious soil erosion, which provides sediments to downstream micro-dams, leading to reductions in the storage volume and lifetime of the micro-dams [3,5].”

to

“The storage volume and lifetime of the micro-dams have been reduced because serious soil erosion provides sediments to downstream micro-dams [3,5].”

We believe that this new sentence adequately addresses the Reviewer’s comment.

Point 4: p. 3 What does this mean: “changeful hydraulic properties”? What does “imprudent vegetation cover” mean?

Response 4: P.3 L.6

In view of the Reviewer’s comment, we have revised the phrase as follows.

1) “by rocks/soils with changeful hydraulic properties”

to

“by different types of rocks/soils”

2) “imprudent vegetation cover”

to

“insufficient vegetation cover by cutting almost all the residues of wheat, teff, and barley, etc.”

We believe that this new information adequately addresses the Reviewer’s comment.

Point 5: p. 4 The photos in this paper are very helpful. Could the authors also include a photo of the Adizaboy dam?

Response 5: P.4 L.1

In the original manuscript, we did not include a photo of the Adizaboy dam. In view of the Reviewer’s comment, we have included a new picture of the Adizaboy micro-dam as Figure 2. Other figure numbers have been changed as well.

Point 6: I do not know what this means: “The farmers trail in the micro-watershed traverses in a north-south direction over the embankment.”

Response 6: P.4 L.7

In view of the Reviewer’s comment, we have included the farmers trail in Figure 6.

Point 7: p. 6-7 The Concepts from the Constructability Implementation Guide should be reformatted as a table. As text, they take up too much room and disrupt the flow of the narrative.

Response 7: P.6 L.25 to P.7 L16

In view of the Reviewer’s comment, we have added a new table describing constructability concepts.

Point 8: p. 7 What does this mean: “Constructability knowledge, such as sediment depth on reclamation farmland of about 20 cm, was obtained from the literature review [10] in terms of cost/benefit ratio”? That onions should grow in a medium with a depth of 20 cm?

Response 8: P.7 L.25

In view of the Reviewer’s comment, we have modified the following sentence.

“Constructability knowledge, such as sediment depth on reclamation farmland of about 20 cm, was obtained from the literature review [10] in terms of cost/benefit ratio.”

to

“We thought that onions should grow with a sediment depth of about 20 cm, which includes the soil loss of 5 cm, in terms of cost/benefit ratio [10].”

Point 9: It is unclear what this means: “Stone bud surrounding reclamation farmland played a role in natural drainage.”

Response 9: P.7 L.35

In view of the Reviewer’s comment, we have modified the following sentence.

“Stone bud surrounding reclamation farmland played a role in natural drainage.”

to

“Abundant irrigation water naturally moved out through the stone bud, which surrounds the perimeter of reclaimed farmland.”

Point 10: What is the Tukey Cramer method?

Response 10: P.7 L.36

In view of the Reviewer’s comment, we have included the following brief explanation after “the Tukey Kramer method.”

“(multiple comparison procedure for statistical analysis)”

Point 11: p. 11 What does this mean: “The design change of the farm pond was conducted to remove the concrete up-downstairs in it and to reduce the use of the framework, sand, and cement,”?

Response 11: P.11 L.20-L.21

In view of the Reviewer’s comment, we tried to modify the following phrase.

“The design change of the farm pond was conducted to remove the concrete up-downstairs in it and to reduce the use of the framework, sand, and cement,”

to

“The wooden ladder was attached in the farm pond as concrete up-downstairs were removed,”

However, in view of the Reviewer 3’s comments, we summarized “Original design \ Concrete up-downstairs \ Modified design \ Wooden ladder” in Table 2.

Reviewer 2 Report

Dear authors. I have read the manuscript of the article with great interest. Unfortunately, the article cannot be published in its current form for a number of reasons.
1. There is no substantial part. What, in fact, was done by you? In the section with Materials and Methods the criteria developed by a specialized Institute are given, but they are given in a generalized form without specifics. How fair are these criteria? There is no answer to this in the manuscript.
2. In the section with results, there are no quantitative assessments. Obviously, the work done by the authors was aimed either at reducing costs, or at increasing fertility, or at obtaining other benefits. Where are these estimates? Has the fertility of soils grown? Did you get a bigger yield? Have the work paid off?
3. The main question is what was the purpose of these works using onions as an example, if they grow on infertile soils. The authors themselves write that onions grow at a shallow depth, then why increase the thickness of the soil?
4. How was sediment collected from the dams? How fertile were these sediments? We need to assess the humusivity, chemical and biological composition of sediments.

5. The authors describe gully erosion in the study area. What about erosion in the catchments themselves? The amount of precipitation should ensure stable flushout from the fields. These questions arise because it seems that there is no more erosion other than gully erosion recorded by the authors.
6. The gullies, judging by the graphical material presented, no longer contain fertile horizons (the talveg of the gullies contains multiple debris from rocks), which could somehow have been washed away for reclamation.What will the authors use in the future when the sediments in the dams disappear?
7. How do the authors propose to control soil flushing?

Author Response

July 17, 2020

To: The Reviewer 2, Sustainability

Dear Sir/Madam;

We would like to submit our revised manuscript entitled “Constructability Criteria for Farmland Reclamation and Vegetable Cultivation by Using Micro-dam Sediments in Tigray, Ethiopia,” which was originally submitted on June 29, 2020, and which has the following manuscript ID “sustainability-865755.”

We thank the Referee 2 for making time available to this paper. We appreciate very much for his/her giving us seven comments, to which we would like to respond as follows.

  1. There is no substantial part. What, in fact, was done by you? In the section with Materials and Methods the criteria developed by a specialized Institute are given, but they are given in a generalized form without specifics. How fair are these criteria? There is no answer to this in the manuscript.

(Response 1) In view of Reviewer 1’s comment, the substantial part of the paper would be section 3.2. We have played a role as described in “Author Contributions,” which all authors agreed. The constructability concepts developed by the Construction Industry Institute are well known and many papers have been published. They are given in a generalized form without specifications. However, experiences and knowledge are considered to be important, and they have been shared by us. In the future, it might be necessary to conduct a questionnaire survey to evaluate the importance of constructability criteria, which might depend on the local conditions.

  1. In the section with results, there are no quantitative assessments. Obviously, the work done by the authors was aimed either at reducing costs, or at increasing fertility, or at obtaining other benefits. Where are these estimates? Has the fertility of soils grown? Did you get a bigger yield? Have the work paid off?

(Response 2) Thank you for your questions. However, the authors state the quantitative information on sediment volume and onion yield in this paper. We got a bigger yield than the national average yield. Cost information has been included in a table in discussion and conclusions. The sediment analysis results have already been published in our paper (https://www.mdpi.com/2071-1050/11/7/2038/htm) as the following table, which shows the sediment fertility is better than the bare land fertility. In the future farmers will take over the reclaimed farmland from the Agulae wareda office. We expect that the work will have paid off probably with the sustained help of support organizations.

  1. The main question is what was the purpose of these works using onions as an example, if they grow on infertile soils. The authors themselves write that onions grow at a shallow depth, then why increase the thickness of the soil?

(Response 3) The purpose of our works is to increase the vegetable cultivation on reclaimed farmland, for which the constructability criteria have been applied, for young landless farmers, etc. as a social implementation method. We did not increase the soil thickness to grow onions because reclaimed farmland was constructed on bare land where the topsoil has been excavated and removed to construct the micro-dam embankment near the reclaimed farmland. In the future, we should include a food value chain analysis.

  1. How was sediment collected from the dams? How fertile were these sediments? We need to assess the humusivity, chemical and biological composition of sediments.

(Response 4) We collected sediments by using a donkey. The sediment analysis results to assess the properties have been shown in the table in reply to question No.2. The sediment has a neutral reaction (pH 7.3 vs pH 8.1 on bare land) and contains a high percentage of organic carbon (2.5% vs 1.8% on bare land), available phosphorus (9.2 mg/kg vs 8.8 mg/kg on bare land) and exchangeable potassium (25 cmol(+)/kg vs 14.1 cmol(+)/kg (9.8 g/kg vs 5.5 g/kg) on bare land).

  1. The authors describe gully erosion in the study area. What about erosion in the catchments themselves? The amount of precipitation should ensure stable flushout from the fields. These questions arise because it seems that there is no more erosion other than gully erosion recorded by the authors.

(Response 5) Rill and sheet erosions have occurred in the field. We focused on gully erosion because it was obviously much larger than them and formed in the downstream of them in the micro-watershed. We have been conducting the wheat cultivation test in the micro-watershed for two years, which includes the measurement of the amount of precipitation and soil erosion in the farmland. The paper will be submitted to Sustainability in the future.

  1. The gullies, judging by the graphical material presented, no longer contain fertile horizons (the talveg of the gullies contains multiple debris from rocks), which could somehow have been washed away for reclamation. What will the authors use in the future when the sediments in the dams disappear?

(Response 6) The gullies have been active. The fine soils and clays have been transported from the micro-watershed to the downstream micro-dam through the gullies. Some of them still might have been on the way to the micro-dam. When we use up the sediments in the micro-dams in the future, it will be able to recover the loss of storage water. After we wait for the sediment accumulation for a while, we might be able to continue the farmland reclamation in the micro-watershed. If the sediments no longer contain the fertile horizons, we have to use the fertilizer to cultivate vegetables. We might have to care about the salt damage in the future in case that farmers continue to use fertilizer.

  1. How do the authors propose to control soil flushing?

(Response 7) Thank you for your questions. We will mitigate soil erosion in the micro-watershed by conducting conservation agriculture. We have been repeating the wheat (staple food) test, which changes the wheat residue, for three years to find the most effective residue method to reduce soil erosion in the field.

Please find the following our point-to-point responses to other reviewers’ comments, and attached revised manuscript with changes shown. We would like to thank the reviewer 2 for their valuable comments, which we think have resulted in an improved manuscript. We trust that the revised manuscript will meet the approval of the editor and reviewers, and that it would be suitable for publication in Sustainability.

Sincerely yours,

Kazuhisa KODA

Senior Researcher

Rural Development Division

Japan International Research Center for Agricultural SciencesOhwashi, Tsukuba, Ibaraki 305-8686, Japan

Phone : +81-29-838-6676 Fax : +81-29-838-6682

E-mail : <kodakazu@affrc.go.jp>

***** Response to Reviewer 1's Comments *****

(Point) The manuscript “Constructability Criteria for Farmland Reclamation and Vegetable Cultivation by Using Micro-dam Sediments in Tigray, Ethiopia” reports on a project to use sediments accumulating in a dam in Ethiopia to enrich eroded farmland upstream from the dam. More than anything, this manuscript reads as a project report, summarizing the initial problem, what was done, and what the outcome was. To me, the most interesting and useful part of the paper is Section 3.2, which applies the general criteria for a successful construction project, to the specific project being discussed.

As written, the manuscript feels somewhat isolated. In essence, it is a case study, but, beyond pointing out that there are many micro-dams in Ethiopia like the Tigray dam, they do not try to extend the findings from this project to other cases. The manuscript presents a helpful account of what happened in one place, but without a lot of explicit focus on broader extensions.

The style of the writing is somewhat idiosyncratic, especially in terms of diction, preposition use, etc. To me, this is generally not a barrier to understanding. I commented on a few lines or phrases that I had trouble understanding. The decision of whether or not to require more editing to conform to standard English-language usage for science writing is up to the editors.

(Response) We would like to express our appreciation to the Reviewer for his or her insightful comments, which have helped us significantly improve the paper “Constructability Criteria for Farmland Reclamation and Vegetable Cultivation by Using Micro-dam Sediments in Tigray, Ethiopia.”

The problem we have challenged in the present paper will be a major issue in the field. We appreciate the Reviewer’s comment. However, we think that we have chosen one of the representative micro-dams. In general, the sedimentation problem in Adizaboy seems to be very similar to that in other micro-dams based on the literature review. Unfortunately, we have not been able to go out of Tigray region due to the instability of the political situation in Ethiopia. Our field activities, which have recently been conducted, included other micro-dams such as Maileba, Gum 30, Shilanat, and Ground Ho in Tigray, etc. We had a meeting with a JICA expert in Ethiopia several times in ADD as well. We were about to apply for extending our training activities in cooperation with JICA and Embassy of Japan in Ethiopia as well as to prepare for the electrical prospecting to specify the water leakage location of Adizaboy micro-dam for the further study just before the spread of COVID-19.

Point 1: p. 2 I do not understand what these figures mean: “The ratio of agricultural land area to total land area accounts for 36%, arable land area 15%, permanent cropland area 1%, and forest area 13% [9].”

Response 1: P.2 L.1

We appreciate the Reviewer’s useful comments on the statistical data. I think that these figures mean that “It will be necessary to conduct agricultural development in Ethiopia,” which has been added after “The ratio of agricultural land area to total land area accounts for 36%, arable land area 15%, permanent cropland area 1%, and forest area 13% [9].”

Point 2: This sentence marks the first time micro-dams are mentioned in the main text: “It has been estimated that 50% of the micro-dams in northern Ethiopia have had their life expectancy declined from 26 to 13 years due to sedimentation [10-11].” The context for understanding what they are isn’t provided until the next paragraph. I would move this sentence and the next one to be with the introduction to micro-dams. It also might be helpful to add a sentence or two to that paragraph explaining exactly what a micro-dam is. (As someone who doesn’t know, I just assumed that they are small dams, but this doesn’t answer other questions, such as whether they produce electricity, whether they are primarily a means of creating a reservoir, etc.)

Response 2: P.2 L.7

Thank you for pointing out this error. In view of the Reviewer’s comment, we have added the following sentence.

“A micro-dam is a small dam or reservoir to store water for domestic, livestock, and irrigation uses.”

The following two sentences have also moved to the next paragraph.

“It has been estimated that 50% of the micro-dams in northern Ethiopia have had their life expectancy declined from 26 to 13 years due to sedimentation [10-11]. Such a process resulted in lower soil nutrients, such as organic carbon (OC), total nitrogen (N), available phosphorus (P), and exchangeable cations in the soils of the watershed than those in micro-dam sediments [10].”

Reference numbers have been changed from 11 to 12 and from 12 to 11.

Point 3: This sentence is unclear to me: “Thus, the goals are more often short-term than expected due to serious soil erosion, which provides sediments to downstream micro-dams, leading to reductions in the storage volume and lifetime of the micro-dams [3,5].”

Response 3: P.2 L.14

In view of the Reviewer’s comment, we have revised the following sentence.

“Thus, the goals are more often short-term than expected due to serious soil erosion, which provides sediments to downstream micro-dams, leading to reductions in the storage volume and lifetime of the micro-dams [3,5].”

to

“The storage volume and lifetime of the micro-dams have been reduced because serious soil erosion provides sediments to downstream micro-dams [3,5].”

We believe that this new sentence adequately addresses the Reviewer’s comment.

Point 4: p. 3 What does this mean: “changeful hydraulic properties”? What does “imprudent vegetation cover” mean?

Response 4: P.3 L.6

In view of the Reviewer’s comment, we have revised the phrase as follows.

“by rocks/soils with changeful hydraulic properties”

to

“by different types of rocks/soils”

“imprudent vegetation cover”

to

“insufficient vegetation cover by cutting almost all the residues of wheat, teff, and barley, etc.”

We believe that this new information adequately addresses the Reviewer’s comment.

Point 5: p. 4 The photos in this paper are very helpful. Could the authors also include a photo of the Adizaboy dam?

Response 5: P.4 L.1

In the original manuscript, we did not include a photo of the Adizaboy dam. In view of the Reviewer’s comment, we have included a new picture of the Adizaboy micro-dam as Figure 2. Other figure numbers have been changed as well.

Point 6: I do not know what this means: “The farmers trail in the micro-watershed traverses in a north-south direction over the embankment.”

Response 6: P.4 L.7

In view of the Reviewer’s comment, we have included the farmers trail in Figure 6.

Point 7: p. 6-7 The Concepts from the Constructability Implementation Guide should be reformatted as a table. As text, they take up too much room and disrupt the flow of the narrative.

Response 7: P.6 L.25 to P.7 L16

In view of the Reviewer’s comment, we have added a new table describing constructability concepts.

Point 8: p. 7 What does this mean: “Constructability knowledge, such as sediment depth on reclamation farmland of about 20 cm, was obtained from the literature review [10] in terms of cost/benefit ratio”? Those onions should grow in a medium with a depth of 20 cm?

Response 8: P.7 L.25

In view of the Reviewer’s comment, we have modified the following sentence.

“Constructability knowledge, such as sediment depth on reclamation farmland of about 20 cm, was obtained from the literature review [10] in terms of cost/benefit ratio.”

to

“We thought that onions should grow with a sediment depth of about 20 cm, which includes the soil loss of 5 cm, in terms of cost/benefit ratio [10].”

Point 9: It is unclear what this means: “Stone bud surrounding reclamation farmland played a role in natural drainage.”

Response 9: P.7 L.35

In view of the Reviewer’s comment, we have modified the following sentence.

“Stone bud surrounding reclamation farmland played a role in natural drainage.”

to

“Abundant irrigation water naturally moved out through the stone bud, which surrounds the perimeter of reclaimed farmland.”

Point 10: What is the Tukey Cramer method?

Response 10: P.7 L.36

In view of the Reviewer’s comment, we have included the following brief explanation after “the Tukey Kramer method.”

“(multiple comparison procedure for statistical analysis)”

Point 11: p. 11 What does this mean: “The design change of the farm pond was conducted to remove the concrete up-downstairs in it and to reduce the use of the framework, sand, and cement,”?

Response 11: P.11 L.20-L.21

In view of the Reviewer’s comment, we tried to modify the following phrase.

“The design change of the farm pond was conducted to remove the concrete up-downstairs in it and to reduce the use of the framework, sand, and cement,”

to

“The wooden ladder was attached in the farm pond as concrete up-downstairs were removed,”

However, in view of the Reviewer 3’s comments, we summarized “Original design Concrete up-downstairs Modified design Wooden ladder” in Table 2.

***** Response to Reviewer 3's Comments *****

(Point) Many photos are not clear. It’s better to indicate the orientation of a picture with letters (like E-W) inside the picture than with N arrows. They are much too early in the text, sometimes several pages before the mention in the text.

The authors have made some descriptions on the “constructability” criteria, which is the focus of this paper. However, I personally think that, by transforming these texts into diagrams or tables, the authors can better present their ideas in a more logical and analytical way.

Not many English mistakes were found in the paper. However, there are some minor mistakes that the authors will need to look at closely. Also, it is suggested that the authors can use semicolon to separate items in one sentence. This will help to make the paper even more readable.

(Response) We would like to express our appreciation to the Reviewer 3 for his or her insightful comments, which have helped us significantly improve the paper.

  • Larger and clear photos have been copied to my paper to show them clearly. The orientation in some pictures was described in Fig. 2 to 5 and 8 to 9.
  • I think that the constructability criteria are the focus of this paper.
  • The table has been used to present the ideas.
  • We have used semicolons to separate items in one sentence.

Point 1: Page 2 (red): Not clear. It is suggested that the authors can perhaps rephrase the sentence and pay attention to the use of punctuation (also, Page 4, marked in red)

Response 1: P.2 L.19-20

In accordance with the Reviewer’s comment, we have revised the sentence as follows.

“In Ethiopian highlands, poor water resources to fulfil water demand is a significant factor raised for yielding agricultural crops.”

to

“In Ethiopian highlands, poor water resources, which should fulfil water demand, are a significant factor to yield agricultural crops.”

Point 2: Fig. 1 (a) & (b): Can the “Remark” be integrated into the descriptions of Fig. 1?

Response 2: P.3 Figure 1 (a) (b)

In accordance with the Reviewer’s comment, we have integrated the following remark of “The black area displays Ethiopia” and “Bold dot line shows the national boundary of Ethiopia, and the black area Tigray” into the caption of Figure 1.

Point 3: Fig. 1(c) + Page 3 (red)

Not clear about “the average slope of the survey line was 8.8%” especially the “survey line” is not clearly marked in the map/Legend. It is also suggested that the authors can perhaps use a topographic map instead to show the altitude of observed sites; and perhaps specify the criteria for selecting the observation points.

Response 3: P.3 L.20 Figure 1 (c)

We appreciate the Reviewer’s comment. However, the representative survey line is described as one of the roads in Figure 1 (c). It has been shown in the map/Legend by changing the line shape. The reviewer 3 has requested the topographic map. We consider that another paper describing the topographic map is now under review. We think that we should not describe it in this paper without permission. For this reason, we would prefer not to include the map in this figure. There was no obstacle to conduct the level survey around the observation points.

Point 4: Page 4 (red) – 1

Not sure about this. 21.4% + 87.1% + 17.6%? What is the 17.6% for?

Response 4: P.4 L.14-16

Thank you for pointing out this error. In accordance with the Reviewer’s comment, we have modified the following sentence.

“About 1.82 km2 of croplands in Adizaboy micro-watershed accounts for 21.4%, and 87.1% of them is situated on a comparatively flat land whose slope is less than 10 degrees and accounts for 17.6% [18].”

to

“The ratio of about 1.82 km2 of croplands to the area in Adizaboy micro-watershed accounts for 21.4%, and 87.1% of them is situated on a comparatively flat land whose slope is less than 10 degrees [18].”

Point 5: Page 4 (red) – 2

Punctuation?

Response 5: P.4 L.25

Thank you for pointing out this error. In accordance with the Reviewer’s comment, we have modified the following words. A punctuation was added between “point” and “water.”

“At this point water inflows through”

to

“At this point, water inflows through”

Point 6: Page 4 (red) – 3

“be incorporated into the gully erosion” or “come into the gully erosion zone/channel”?

Response 6: P.4 L.26

Thank you for pointing out this error. In accordance with the Reviewer’s comment, we have modified the following words.

“come into the gully erosion”

to

“come into the gully erosion zone/channel”

Point 7: Page 4 (red) – 4

Stones

Response 7: P.4 L.34

This is a mistake. In accordance with the Reviewer’s comment, we have modified the following words.

“stone”

to

“stones”

Point 8: Page 5 (red)

“in the upstream” and “in mid-stream Adizaboy micro-watershed”? Please check the grammar.

Response 8: P.5 L.3 and 5

We apologize for the errors. In accordance with the Reviewer’s comment, we have modified the following words.

1) “at mid-stream Adizaboy micro-watershed”

to

”in mid-stream Adizaboy micro-watershed”

2) “at midstream”

to

“in the midstream”

3) “at upstream”

to

“in the upstream”

Point 9: Fig. 4

“mid-stream” or “midstream”?

Response 9: P.5 L.8

We apologize for the errors. In view of the Reviewer’s comment, we have modified the following words in Figure 4.

“at midstream Adizaboy micro-watershed”

to

“in mid-stream Adizaboy micro-watershed”

Point 10: Page 6 (red) – 1

“in March and April” or “from March to April”?

Response 10: P.6 L.3

This expression is incorrect. In view of the Reviewer’s comment, we have modified the following words.

“in March to April”

to

“in March and April.”

Point 11: Page 6 (red) – 2

“disappeared”.

Response 11: P.6 L.4

Thank you for pointing out this error. In accordance with the Reviewer’s comment, we have modified the following words.

“was disappeared”

to

“disappeared”

Point 12: Page 6 (red) – 3

“with a height of 0.5 m and width of 0.6 m” or “0.5 m high and 0.6 m wide”.

Response 12:  P.6 L.7

This expression is incorrect. In accordance with the Reviewer’s comment, we have modified the following words.

“height 0.5 m and width 0.6 m”

to

“with a height of 0.5 m and width of 0.6 m”

Point 13: Page 6/7 (red)

It is suggested that the authors can perhaps transform the descriptions of the seven constructability concepts into a table or flowchart.

Response 13:  P.6 L.25-P.7 L.16

In accordance with the Reviewer’s comment, seven constructability concepts have been described in Table 1.

Point 14: Fig. 7

The text marked in red has been partially omitted.

Response 14:  P.8 Figure 7

We appreciate the Reviewer’s comment. In accordance with the Reviewer’s comment, “Irrigation method” is fully shown in the axis label. “method” has been added to each item in Figure 7.

Point 15: Page 8 (red)

It is suggested that the authors can use the semicolon to separate the items. The same technique can also be applied in other parts of this paper.

Response 15:  P.8 L.4-8

In accordance with the Reviewer’s comment, the semicolon has been additionally used to separate items. “,” has been replaced by “;” in P.6 L6-9; P.9 L.28-34; P.10 L15-20; and P.10 L.25-29 as well.

Point 16: Page 11

Will be interesting to see the information described in a table.

Response 16:  P.11 L.20-30

In accordance with the Reviewer’s comment, the information has been described in Table 2.

Reviewer 3 Report

General comment:

  1. Many photos are not clear. It’s better to indicate the orientation of a picture with letters (like E-W) inside the picture than with N arrows. They are much too early in the text, sometimes several pages before the mention in the text.
  2. The authors have made some descriptions on the “constructability” criteria, which is the focus of this paper. However, I personally think that, by transforming these texts into diagrams or tables, the authors can better present their ideas in a more logical and analytical way.
  3. Not many English mistakes were found in the paper. However, there are some minor mistakes that the authors will need to look at closely. Also, it is suggested that the authors can use semicolon to separate items in one sentence. This will help to make the paper even more readable.

Page 2 (red): Not clear. It is suggested that the authors can perhaps rephrase the sentence and pay attention to the use of punctuation (also, Page 4, marked in red)

Fig. 1 (a) & (b): Can the “Remark” be integrated into the descriptions of Fig. 1?

Fig. 1(c) + Page 3 (red): Not clear about “the average slope of the survey line was 8.8%” especially the “survey line” is not clearly marked in the map/Legend. It is also suggested that the authors can perhaps use a topographic map instead to show the altitude of observed sites; and perhaps specify the criteria for selecting the observation points.

Page 4 (red) - 1: Not sure about this. 21.4% + 87.1% + 17.6%? What is the 17.6% for?

Page 4 (red) – 2: Punctuation?

Page 4 (red) – 3: “be incorporated into the gully erosion” or “come into the gully erosion zone/channel”?

Page 4 (red) – 4: Stones

Page 5 (red): “in the upstream” and “in mid-stream Adizaboy micro-watershed”? Please check the grammar.

Fig. 4: “mid-stream” or “midstream”?

Page 6 (red) - 1: “in March and April” or “from March to April”?

Page 6 (red) – 2: “disappeared”.

Page 6 (red) – 3: “with a height of 0.5 m and width of 0.6 m” or “0.5 m high and 0.6 m wide”.

Page 6/7 (red): It is suggested that the authors can perhaps transform the descriptions of the seven constructability concepts into a table or flowchart.

Fig. 7: The text marked in red has been partially omitted.

Page 8 (red): It is suggested that the authors can use the semicolon to separate the items. The same technique can also be applied in other parts of this paper.

Page 11: Will be interesting to see the information described in a table.

Author Response

July 17, 2020

To: The Reviewer 3, Sustainability

Dear Sir/Madam;

We would like to submit our revised manuscript entitled “Constructability Criteria for Farmland Reclamation and Vegetable Cultivation by Using Micro-dam Sediments in Tigray, Ethiopia,” which was originally submitted on June 29, 2020, and which has the following manuscript ID “sustainability-865755.”

We thank the Reviewer 3 for making time available to this paper. Please find the following our point-to-point responses to the reviewer 3’s comments, and attached revised manuscript with changes shown. We would like to thank the reviewer 3 for their valuable comments, which we think have resulted in an improved manuscript. We trust that the revised manuscript will meet the approval of the editor and reviewers, and that it would be suitable for publication in Sustainability.

Sincerely yours,

Kazuhisa KODA

Senior Researcher

Rural Development Division

Japan International Research Center for Agricultural Sciences

Ohwashi, Tsukuba, Ibaraki 305-8686, Japan

Phone : +81-29-838-6676 Fax : +81-29-838-6682

E-mail : <kodakazu@affrc.go.jp>

***** Response to Reviewer 3's Comments *****

(Point) Many photos are not clear. It’s better to indicate the orientation of a picture with letters (like E-W) inside the picture than with N arrows. They are much too early in the text, sometimes several pages before the mention in the text.

The authors have made some descriptions on the “constructability” criteria, which is the focus of this paper. However, I personally think that, by transforming these texts into diagrams or tables, the authors can better present their ideas in a more logical and analytical way.

Not many English mistakes were found in the paper. However, there are some minor mistakes that the authors will need to look at closely. Also, it is suggested that the authors can use semicolon to separate items in one sentence. This will help to make the paper even more readable.

(Response) We would like to express our appreciation to the Reviewer 3 for his or her insightful comments, which have helped us significantly improve the paper.

  • Larger and clear photos have been copied to my paper to show them clearly. The orientation in some pictures was described in Fig. 2 to 5 and 8 to 9.
  • I think that the constructability criteria are the focus of this paper.
  • The table has been used to present the ideas.
  • We have used semicolons to separate items in one sentence.

Point 1: Page 2 (red): Not clear. It is suggested that the authors can perhaps rephrase the sentence and pay attention to the use of punctuation (also, Page 4, marked in red)

Response 1: P.2 L.19-20

In accordance with the Reviewer’s comment, we have revised the sentence as follows.

“In Ethiopian highlands, poor water resources to fulfil water demand is a significant factor raised for yielding agricultural crops.”

to

“In Ethiopian highlands, poor water resources, which should fulfil water demand, are a significant factor to yield agricultural crops.”

Point 2: Fig. 1 (a) & (b): Can the “Remark” be integrated into the descriptions of Fig. 1?

Response 2: P.3 Figure 1 (a) (b)

In accordance with the Reviewer’s comment, we have integrated the following remark of “The black area displays Ethiopia” and “Bold dot line shows the national boundary of Ethiopia, and the black area Tigray” into the caption of Figure 1.

Point 3: Fig. 1(c) + Page 3 (red)

Not clear about “the average slope of the survey line was 8.8%” especially the “survey line” is not clearly marked in the map/Legend. It is also suggested that the authors can perhaps use a topographic map instead to show the altitude of observed sites; and perhaps specify the criteria for selecting the observation points.

Response 3: P.3 L.20 Figure 1 (c)

We appreciate the Reviewer’s comment. However, the representative survey line is described as one of the roads in Figure 1 (c). It has been shown in the map/Legend by changing the line shape. The reviewer 3 has requested the topographic map. We consider that another paper describing the topographic map is now under review. We think that we should not describe it in this paper without permission. For this reason, we would prefer not to include the map in this figure. There was no obstacle to conduct the level survey around the observation points.

Point 4: Page 4 (red) – 1

Not sure about this. 21.4% + 87.1% + 17.6%? What is the 17.6% for?

Response 4: P.4 L.14-16

Thank you for pointing out this error. In accordance with the Reviewer’s comment, we have modified the following sentence.

“About 1.82 km2 of croplands in Adizaboy micro-watershed accounts for 21.4%, and 87.1% of them is situated on a comparatively flat land whose slope is less than 10 degrees and accounts for 17.6% [18].”

to

“The ratio of about 1.82 km2 of croplands to the area in Adizaboy micro-watershed accounts for 21.4%, and 87.1% of them is situated on a comparatively flat land whose slope is less than 10 degrees [18].”

Point 5: Page 4 (red) – 2

Punctuation?

Response 5: P.4 L.25

Thank you for pointing out this error. In accordance with the Reviewer’s comment, we have modified the following words. A punctuation was added between “point” and “water.”

“At this point water inflows through”

to

“At this point, water inflows through”

Point 6: Page 4 (red) – 3

“be incorporated into the gully erosion” or “come into the gully erosion zone/channel”?

Response 6: P.4 L.26

Thank you for pointing out this error. In accordance with the Reviewer’s comment, we have modified the following words.

“come into the gully erosion”

to

“come into the gully erosion zone/channel”

Point 7: Page 4 (red) – 4

Stones

Response 7: P.4 L.34

This is a mistake. In accordance with the Reviewer’s comment, we have modified the following words.

“stone”

to

“stones”

Point 8: Page 5 (red)

“in the upstream” and “in mid-stream Adizaboy micro-watershed”? Please check the grammar.

Response 8: P.5 L.3 and 5

We apologize for the errors. In accordance with the Reviewer’s comment, we have modified the following words.

1) “at mid-stream Adizaboy micro-watershed”

to

”in mid-stream Adizaboy micro-watershed”

2) “at midstream”

to

“in the midstream”

3) “at upstream”

to

“in the upstream”

Point 9: Fig. 4

“mid-stream” or “midstream”?

Response 9: P.5 L.8

We apologize for the errors. In view of the Reviewer’s comment, we have modified the following words in Figure 4.

“at midstream Adizaboy micro-watershed”

to

“in mid-stream Adizaboy micro-watershed”

Point 10: Page 6 (red) – 1

“in March and April” or “from March to April”?

Response 10: P.6 L.3

This expression is incorrect. In view of the Reviewer’s comment, we have modified the following words.

“in March to April”

to

“in March and April.”

Point 11: Page 6 (red) – 2

“disappeared”.

Response 11: P.6 L.4

Thank you for pointing out this error. In accordance with the Reviewer’s comment, we have modified the following words.

“was disappeared”

to

“disappeared”

Point 12: Page 6 (red) – 3

“with a height of 0.5 m and width of 0.6 m” or “0.5 m high and 0.6 m wide”.

Response 12:  P.6 L.7

This expression is incorrect. In accordance with the Reviewer’s comment, we have modified the following words.

“height 0.5 m and width 0.6 m”

to

“with a height of 0.5 m and width of 0.6 m”

Point 13: Page 6/7 (red)

It is suggested that the authors can perhaps transform the descriptions of the seven constructability concepts into a table or flowchart.

Response 13:  P.6 L.25-P.7 L.16

In accordance with the Reviewer’s comment, seven constructability concepts have been described in Table 1.

Point 14: Fig. 7

The text marked in red has been partially omitted.

Response 14:  P.8 Figure 7

We appreciate the Reviewer’s comment. In accordance with the Reviewer’s comment, “Irrigation method” is fully shown in the axis label. “method” has been added to each item in Figure 7.

Point 15: Page 8 (red)

It is suggested that the authors can use the semicolon to separate the items. The same technique can also be applied in other parts of this paper.

Response 15:  P.8 L.4-8

In accordance with the Reviewer’s comment, the semicolon has been additionally used to separate items. “,” has been replaced by “;” in P.6 L6-9; P.9 L.28-34; P.10 L15-20; and P.10 L.25-29 as well.

Point 16: Page 11

Will be interesting to see the information described in a table.

Response 16:  P.11 L.20-30

In accordance with the Reviewer’s comment, the information has been described in Table 2.

Round 2

Reviewer 2 Report

Dear authors. Thank you for answering the questions. Some of the comments of all reviewers were taken into account in the revised version of the manuscript. However, I would like the methodological part of the work to be disclosed in more detail, or an appropriate reference to other work by the authors.
I wish the authors success in their further promotion of the manuscript after the final editorial revision.

Author Response

Dear The Reviewer 2,

We are submitting a revised version of our manuscript (Ref No: 865755), which was submitted on July 17, 2020. It has been revised in line with the Reviewer 2's comments of "an appropriate reference to other work by authors." The main change is as follows.

P.6 L.3 We have added one citation “Koda, K.; Girmay, G.; Berihu, T.; Nagumo, F. Reservoir Conservation in a Micro-Watershed in Tigray, Ethiopian Highlands. Sustainability. 2019, 11, doi:10.3390/su11072038" as [19] in the references.

We also made some other corrections mainly in Abstract and Keywords. Please find attached a revised copy of the manuscript with changes shown in red. We trust that the revised manuscript will meet the approval of the editor and the reviewer 2. We very much appreciate the helpful comments and suggestions made by the reviewers.

Sincerely yours,

Kazuhisa KODA   

Reviewer 3 Report

I have well-received the second review of “Constructability Criteria for Farmland Reclamation and Vegetable Cultivation by Using Micro-dam Sediments in Tigray, Ethiopia” by Dr. Koda et al,. The authors replied all my questions and made corrections related thereto. I really appreciate their efforts for making all of these modifications based on my feedback.
This paper presents the constructability criteria to reclaim farmlands to solve the micro-dam soil sedimentation, shows the feasibility of vegetable cultivation on reclaimed farmland by using sediments in the micro-dam. This is undeniable an interesting paper as the authors attempt to solve the micro-dam soil sedimentation, shows the feasibility of vegetable cultivation on reclaimed farmland.
However, I would recommend the authors to carefully review the format of this paper as a whole, as minor errors of the spacing (between numbers; number and unit; and word and punctuation etc.) and font can still be found in the paper. It is therefore recommended for acceptance on condition of minor revisions.

Author Response

Dear The Reviewer 3,

We are submitting a revised version of our manuscript (Ref No: 865755), which was submitted on July 17, 2020. It has been revised in line with the Reviewer 3's comments of "carefully review the format of this paper as a whole, as minor errors of the spacing (between numbers; number and unit; and word and punctuation etc.) and font." We apologize for the errors and state that the necessary corrections have been made.

The main changes are as follows.

  • P.1 L.31 There is no space between numbers and “%” based on the APA manual. There has not been any modification of this matter.
  • P.1 L.36 A hyphen has been deleted. “130-ton ha-1 yaer-1” has been revised to “130 ton ha-1 year-1.”
  • P.2 L.10 An em dash has been deleted. “—despite” has been revised to “despite.”
  • P.2 L.16 A punctuation has been added. “In Ethiopian highlands” has been revised to “In Ethiopian highlands,”.
  • P.2 L.19 “every” has been added between “in” and “2.4 years.”
  • P.2 L.48 A lengthy word has been deleted. “security/livelihood” has been revised to ”security.”
  • P.3 L.14-15 A punctuation has been added between numbers. “1973-1974 and 1984-1985” has been modified to “1973-1974, and 1984-1985.”
  • P.3 L.18 “above sea level” has been revised to “a.s.l.”
  • P.4 L.2 “up-land” has been revised to “upland.”
  • P.4 L.14 “up- and middle- stream” has been modified to “upstream and middle reaches.”
  • P.4 L.30 all of “up-stream” have been revised from here forward to “upstream.”
  • P.5 L.9 Font of “2) Mid-stream” has been revised from “Time New Roman” to “Palatino Linotype.”
  • P.5 L.8-L.14 All of “mid-stream” have been modified to “middle reaches.”
  • P.7 L.26 A hyphen has been added between words. “soil and water conserving” has been revised to “soil- and water-conserving.”
  • P.7 L.37 “twelve” has been revised to “12” as the number is more than 10.
  • P.7 L.44 A punctuation has been added. “analysis) the onion” has been revised to “analysis), the onion.”
  • P.8 L.18 “After water flow” has been revised to “After water flows.”
  • P.9 L.9 A punctuation has been added. “and reclaimed farmlands should” has been revised to “and reclaimed farmlands, should.”

We also made some other corrections mainly in Abstract and Keywords. Please find attached a revised copy of the manuscript with changes shown in red. We trust that the revised manuscript will meet with the approval of the editor and the reviewer 3. We very much appreciate the helpful comments and suggestions made by the reviewers.

Sincerely yours,

Kazuhisa KODA  
